# Amelioration of Serum Aβ Levels and Cognitive Impairment in APPPS1 Transgenic Mice Following Symbiotic Administration

**DOI:** 10.3390/nu16152381

**Published:** 2024-07-23

**Authors:** Chiara Traini, Irene Bulli, Giorgia Sarti, Fabio Morecchiato, Marco Coppi, Gian Maria Rossolini, Vincenzo Di Pilato, Maria Giuliana Vannucchi

**Affiliations:** 1Histology and Embryology Research Unit, Department of Experimental and Clinical Medicine, University of Florence, 50139 Florence, Italy; chiara.traini@unifi.it (C.T.); irene.bulli@unifi.it (I.B.); giorgia.sarti@unifi.it (G.S.); 2Microbiology and Virology Unit, Deparment of Experimental and Clinical Medicine, University of Florence, 50139 Florence, Italy; fabio.morecchiato@unifi.it (F.M.); marco.coppi@unifi.it (M.C.); gianmaria.rossolini@unifi.it (G.M.R.); 3UO Microbiologia, IRCC Ospedale Policlinico San Martino, Deaprtment of Surgical Science and Integrated Diagnostics (DISC), University of Genoa, 16132 Genoa, Italy; vincenzo.dipilato@unige.it

**Keywords:** Alzheimer’s disease, Akkermansia, intestinal epithelium, microbiota, mucus secretion, neurodegeneration, nutraceuticals

## Abstract

Alzheimer’s disease (AD) is a neurodegenerative process responsible for almost 70% of all cases of dementia. The clinical signs consist in progressive and irreversible loss of memory, cognitive, and behavioral functions. The main histopathological hallmark is the accumulation of amyloid-ß (Aß) peptide fibrils in the brain. To date, the origin of Aß has not been determined. Recent studies have shown that the gut microbiota produces Aß, and dysbiotic states have been identified in AD patients and animal models of AD. Starting from the hypothesis that maintaining or restoring the microbiota’s eubiosis is essential to control Aß’s production and deposition in the brain, we used a mixture of probiotics and prebiotics (symbiotic) to treat APPPS1 male and female mice, an animal model of AD, from 2 to 8 months of age and evaluated their cognitive performances, mucus secretion, Aβ serum concentration, and microbiota composition. The results showed that the treatment was able to prevent the memory deficits, the reduced mucus secretion, the increased Aβ blood levels, and the imbalance in the gut microbiota found in APPPS1 mice. The present study demonstrates that the gut–brain axis plays a critical role in the genesis of cognitive impairment, and that modulation of the gut microbiota can ameliorate AD’s symptomatology.

## 1. Introduction

Alzheimer’s disease (AD) is the most common neurodegenerative disease, accounting for almost 70% of all cases of dementia. AD has been defined by the World Health Organization (WHO) and by Alzheimer’s Disease International as a global public health priority. The WHO Global Data Action Plan 2017–2025 indicates that, in 2015, dementia affected 47 million people worldwide, and this number is expected to increase to 75 million by 2030 and 132 million by 2050, while the economic burden estimate is over USD 1 trillion per year, with progressive increases (https://www.who.int/publications/i/item/global-action-plan-on-the-public-health-response-to-dementia-2017---2025, accessed on 8 March 2017).

AD’s clinical signs consist in a progressive and irreversible loss of memory, and in a decline in cognitive, behavioral, and functional activities that deeply compromises the daily-life performances up to immobility and apathy [1]. The main histopathological hallmark of AD is the extracellular accumulation of misfolded amyloid-ß peptide (Aß) fibrils in the brain [2,3]. To date, the origin of Aß has not been clearly established. Several studies have focused on Aß produced in the brain [3]. However, biogenesis of Aβ in the brain is not increased in sporadic and late-onset AD, the most common phenotype of the disease, while evidence suggests an increase in blood-to-brain delivery [4]. More recent studies have shown that the gut microbiota (MB) and enterocytes produce Aß [5,6], and that the plasma concentration of soluble Aβ is correlated with the cerebral Aβ load in both healthy and Alzheimer’s patients [5,7].

The gut MB consists of almost 10^14^ microorganisms, most of which are bacteria belonging to two main phyla: *Firmicutes* (Gram-positive) and *Bacteroidetes* (Gram-negative). The MB, the gut, and the brain are physiologically and pathologically linked through a “two-way” axis, and the health of each component and of the entire organism depends on their mutual interrelationships [8,9]. The MB and gut produce molecules that are necessary for the correct development and functioning of the brain [10,11], while qualitative and quantitative alterations affecting the MB (i.e., dysbiosis) have been associated with the onset of various diseases of the enteric and central nervous system, including motility disorders, behavioral disorders, neurodegenerative diseases, cerebrovascular accidents, and neuroimmune disorders [7,8,9,11,12,13,14]. In turn, brain diseases alter the neurochemistry of the enteric nervous system, the functioning of the local immune system (IS), and the MB itself [12,13].

Dysbiotic states have been identified in patients affected by AD, and in animal models of the disease. In particular, in the APPPS1 strain, an increased relative abundance of bacterial strains that are large producers of Aβ, such as *B. subtilis* and *E. coli*, was reported [15]. *E. choli* also produces an endotoxin that promotes the formation of Aβ fibrils. In parallel, a decrease in Gram+ microbes was described, mostly represented by Lactobacilli that produce trophic agents for both the gut and brain, such as short-chain fatty acids (SCFAs) and tryptophan [6,8,9,11,16,17,18,19,20].

Starting from the hypothesis that maintaining or restoring the eubiosis of the gut MB is essential to control the production of Aß and avoid its deposition in the brain, in the last decade several studies have investigated the effects of prebiotic or probiotic supplementation on cognitive functions and brain damage in animals and humans affected by AD; the results are encouraging [14,21,22,23,24,25,26]. Nevertheless, although the gut is considered to be both the bridge and the active intermediary between the MB and the brain, studies on this organ are few. Furthermore, there is a lack of studies in which prebiotics and probiotics (symbiotic) were administered together for several months starting from an early age.

In the present study, APPPS1 transgenic (Tg) mice for AD were treated daily with a mixture of pre- and probiotics for 6 months, starting from 2 months of age, when the brain is devoid of measurable histopathological signs of Aβ deposition. Both nutraceuticals were added to the diet, thus avoiding stress-related gavage administration. The aims of this study were as follows: (i) to evaluate the cognitive performances during (at 4 months of age) and at the end of the treatment (at 8 months of age), when this strain is reported to accumulate significant and increasing deposits of Aβ in the brain [27]; (ii) to quantify the mucus secretion in the gut epithelium; (iii) to estimate the Aβ in the serum; and (iv) to examine changes in the gut MB’s composition and structure before the beginning and at the end of the treatment.

## 2. Materials and Methods

### 2.1. Animals

Tg heterozygous APPPS1 (Tg) male and female mice with a C57BL/6J genetic background (B6.Cg-Tg(Thy1-APPSw,Thy1-PSEN1*L166P)21Jckr), along with their hybrid wild-type male and female littermate mice (wt) of the same age, were purchased from Papiling GmbH (Rottemburgh, Germany). The Tg mouse model was generated by Prof. Mathias Jucker (Hertie Institute for Clinical Brain Research, Tübingen, Germany).

The Tg mice express a double mutation, specifically, the mutated human APP gene (Swedish mutation, KM670/672NL) and the mutated human PS1 gene (presenilin 1, L166P), both controlled by the same promoter Thy-1 [27,28].

The animals were housed in an air-conditioned room (temperature 21 °C ± 2 °C), under a 12 h light/dark cycle (lights on from 07:00 h to 19:00 h), with free access to food and water. All experimental protocols were carried out in accordance with the European Communities Council Directive 2010/63/UE and approved by the Italian Ministry of Health (code: 53/2022).

The experimental protocol included the following experimental groups:Tg mice and Wt littermates fed with a standard diet, identified as Tg and Wt, respectively;Tg mice and Wt littermates fed with the standard diet enriched in prebiotics and probiotics, identified as Tg-T and Wt-T, respectively.

Each experimental group consisted of 8–16 animals/age, balanced for sex.

### 2.2. Drugs and Treatment

The treatment consisted in an enrichment of the daily diet with prebiotics and probiotics; it lasted for 6 months, from the end of the 2nd month to the end of the 8th month of life.

### 2.3. Prebiotic Formulation and Administration

The prebiotic was a multi-extract of fibers and plant complexes, containing inulin/FOS (fruit oligosaccharides), kindly provided by Aboca S.P.A. (San Sepolcro, Arezzo, Italy). The chosen dosage was 50 mg inulin/FOS/g diet, corresponding to a 5% increase in inulin/FOS with respect to the standard diet. This percentage is consistent with the literature data [29] and with the dosage commonly suggested as a dietary supplement for humans. The preparation of the pellets containing the mix of the standard diet and prebiotics retained the taste of the standard diet and was prepared by Mucedula s.r.l. (Milan, Italy). The food intake was checked daily for the entire period of administration, weighing the leftovers and replenishing the food pellets.

### 2.4. Probiotic Formulation and Administration

The probiotic consisted of a mixture of 50% each of *Lactobacillus rhamnosus* IMC 501 and *Lactobacillus paracasei* IMC 502, and it was kindly provided by Synbiotec srl (Camerino, Macerata, Italy). The probiotic formulation was supplied without excipients or prebiotics, with a bacterial load corresponding to 100 billion cells per g. The chosen dosage was 1.8 × 10^8^ CFU/day/25 g per mouse. It was calculated following the normalization method with respect to the body’s surface area and considering the recommended human dose (15 × 10^9^ CFU*2/day). The dosage was consistent with the literature [30]. As demonstrated by Verdenelli et al. [31], the chosen bacterial mixture survives at gastric pH, tolerates bile acid, does not generate any side effects, has high adhesion and colonization ability, and is recovered in colonic fecal samples. The viability of the bacteria at room temperature is guaranteed up to 12 h. The probiotics were dispersed in drinking water gelled by adding an instant thickener powder made from cornstarch (2.5 g/100 mL) that is tasteless, devoid of toxicity, and commonly used for swallowing deficiencies in humans. This mode of administration is stress-free for the animal and guarantees a uniform redistribution of the probiotics in the water. The volume of gelled water containing the probiotics was prepared and supplied fresh every day for the entire period of treatment. The gelled water was placed into a Nombrero (Animal Specialties and Provisions, LLC, Quakertown, PA, USA), a container specifically designed to feed rodents with wet food that has proven to be particularly suitable for containing gelled water. The presence of a hook made it possible to hang the Nombrero in the upper part of the cage, so that it would not tip over and the gelled water would not come into contact with the litter in the cage, limiting contamination. However, the basal part, containing the gelled water, was shallow enough to be easily accessible to the mice. The amount of water taken was assessed every day by weighing the residual water. The probiotic concentration was adjusted to the mice’s weight gain and the amount of water consumed. As rodents drink mainly during the night, the water containing the probiotics was administered every evening, and the following morning the residual water was weighed.

To guarantee similar environmental conditions, all of the animals received gelled drinking water, with or without probiotics.

### 2.5. Fecal Collection

Fecal pellets from each mouse were collected the day before starting the diet enriched in prebiotics and probiotics (T0), at the 4th month of age, after 60 days of treatment (T60), and at the 8th month of age, after 180 days of treatment (T180). Animals were placed in individual cages provided with a basal grid for 4 h. The bottom of the cage, where the fecal pellets fell, was covered with blotting paper to absorb the urine, so as to limit the possible fecal contamination. In the end, the mouse returned to its home cage and the fecal pellets were counted, weighed, and stored at −80 °C.

### 2.6. Profiling of the Gut Microbiota

The collected fecal pellets (50 mg) were processed for total DNA extraction using the DNeasy PowerLyzer PowerSoil Kit (Qiagen, Hilden, Germany). Next-generation sequencing of 16S rRNA amplicons (V3–V4 regions) was performed using the Illumina NovaSeq platform, following a 2 × 250 bp paired-end approach. The sequencing results were analyzed using the QIIME (Quantitative Insights Into Microbial Ecology) 2 suite [32]. Briefly, following denoising of raw reads (i.e., error correction, removal of chimeric and singleton sequences, joining of denoised paired-end reads) using the Divisive Amplicon Denoising Algorithm 2 (DADA2) [Callahan] [33], amplicon sequence variants (ASVs) were inferred for each sample. Taxonomic classification of dereplicated ASVs was carried out using a naïve Bayes classifier trained on the SILVA 16S reference database v.138 (https://www.arb-silva.de, accessed on 19 December 2020). Community profiling was evaluated by estimating alpha- and beta-diversity using specific tools implemented in the QIIME2 pipeline. The alpha-diversity, which summarizes the distribution of species abundances in each sample considering species richness and evenness, was evaluated through different diversity indices (e.g., Shannon, Chao1). The beta-diversity, which quantifies (dis)similarities between samples, was evaluated through principal coordinates analysis (PCoA) using phylogenetic (e.g., UniFrac) and non-phylogenetic (e.g., Bray–Curtis) metrics. The 16S rRNA sequence data were deposited in the NCBI Sequence Read Archive (SRA) under the BioProject accession number PRJNA1114948.

### 2.7. Behavioral Tests

Cognitive functions were assessed using the following behavioral tests: rotarod, nesting, and novel object recognition, which were applied at 4 and 8 months of age; marble-burying test, step-down inhibitory avoidance, and Barnes maze, applied only at the end of treatment. The operators (C.T. and I.B.) performed the tests while blinded to the group of origin of the tested animal.

### 2.8. Rotarod

The rotarod test is a performance test based on a rotating rod with forced motor activity. The test evaluates the balance, grip strength, and neuromuscular coordination of the mice. The apparatus consists of a motorized, circular rod turning at a constant and settable speed. The rodents naturally tried to walk on the rotarod to avoid falling on the ground. The task procedure was applied for three consecutive days, and it consisted of two trials (3 min/ trial; intertrial interval of 3 min) per day. The first trial was performed at the lowest speed (15 rpm); the second trial was performed at the highest speed (30 rpm). Each time the mouse lost its balance and fell onto the underlying platform, the rodent was relocated on the rod, until the end of the trial. The latency of the first fall and the number of falls during the first and the second trials were recorded. Following each trial, the apparatus was carefully cleaned with Virkon.

### 2.9. Marble Burying

The marble test is a useful model for detecting anxiety-like behavior. When located in a cage with marbles, mice tend to dig the objects. To perform the test, the cages were filled with bedding material up to a height of 4 cm from the cage floor; 20 glass marbles (5 × 4) were aligned at specific intervals, approximately 2 cm from each other. The mouse was placed in the cage for 30 min, to explore and eventually dig the marbles. In the end, the mouse was carefully returned to its home cage, so that the bedding material was not garbled. The number of marbles that remained unburied and, consequently, the number of marbles buried were used as evaluation parameters.

### 2.10. Nesting

Nesting is a natural behavior of rodents, important for thermoregulation, reproduction, and shelter. Animals were placed in individual cages filled with 0.5 cm of normal bedding and small cylinders (2 cm) of cotton (Cocoon) as nesting material. After 24 h, each nest was photographed, and its quality was assessed on a five-point scale [34]. A largely untouched nest with over 90% of material still intact had a score of 1; a partially torn-up nest, with 50–90% intact, had a score of 2; when less than 50% of the material was intact, without an identifiable nest site, a score of 3 was given; an identifiable, flat nest had a score of 4; a nest with a nearly perfect crater, with walls higher than the animal’s body height on 50% of the circumference, had a score of 5.

### 2.11. Novel Object Recognition (NOR)

The NOR task evaluates rodents’ ability to recognize a novel object in the environment. This methodology assesses the natural preference for novel objects displayed by rodents. NOR trials were conducted in square test boxes (arena 40 × 40 × 35 cm). These consisted of a white plastic box with a removable basal grid over a closet, so that it could be properly cleaned after each trial. The objects used were small plastic toys with a cylindrical or pyramidal shape. A camera mounted above the open field recorded the movements of the mouse throughout the trial. The task procedure consisted of three phases: habituation, familiarization, and testing. During the habituation phase, each mouse was allowed to freely explore the empty open-field arena for 5 min. The animal was then removed from the arena and placed in its holding cage. The following day, during the familiarization phase, each mouse was placed in the open-field arena containing two identical objects (A + A), allowing free exploration of the objects for 10 min. To prevent coercion to explore the objects, the rodents were released in a corner of the arena with their back to the objects. The corner was the same for each animal. On the 3rd day, during the test phase, the animal was returned to the open-field arena enriched with two objects; one was identical to those of the previous day, and the other was novel (A + B). The mouse was allowed free exploration of the objects for 10 min. The new object differed from the previous one only in shape, not in color. After each trial, the objects, the basal grill, and the box wall were cleaned carefully with Virkon. Object interaction for the novel object was calculated as follows: novel object interaction time × 100/sample object interaction time + novel object interaction time. The discrimination index was calculated as follows: novel object interaction time—sample object interaction time/sample object interaction time + novel object interaction time. Mice with less than 20 s of total object interaction during the familiarization were excluded from the analysis [35].

### 2.12. Barnes Maze

The Barnes maze is a dry-land-based rodent behavioral test for assessing spatial learning and memory. This test was conducted according to the previously described method [36]. The Barnes maze setup consisted of an elevated circular platform (122 cm diameter) with 20 equally spaced (5 cm) holes around the perimeter; under one hole there was a dark tunnel for escape, while the remaining 19 holes were false. The mice learned the relationship between cues in the surrounding environment and a fixed escape location. The task procedure consisted of two phases: a training and a probe phase. The training phase consisted of two daily acquisition trials (3 min/trial; intertrial interval ~1 h), repeated for 10 days. Each training phase started by locating the mouse in the center of the platform, under a small dark box. Then, the box was lifted, freeing the mouse to explore the platform. Both bright light and open elevated platforms are aversive to rodents, thus inducing the exploration behavior. The trial was concluded when the mouse entered the escape tunnel or 3 min elapsed. If a mouse failed to find the escape tunnel within the 3 min trial, it was placed in the escape box by the researcher and allowed to stay there for 15 s before removal. The animal was then brought into its holding cage. Following each trial, the maze and escape tunnel were carefully cleaned with Virkon. The probe phase was performed on day 3 after the final session of the training phase and consisted of letting the mouse explore the platform for 1 min. A camera positioned over the platform recorded the performance of each mouse.

### 2.13. Step-Down Inhibitory Avoidance

The inhibitory avoidance apparatus consisted of an open-field Plexiglas box (40 × 40 cm) with a steel rod floor and a Plexiglas platform (4 × 4 × 4 cm) set in the center of the grid floor. Intermittent electric shocks (20 mA, 50 Hz) were delivered to the grid floor by an isolated stimulator. The mice learned to associate the innate stepping down from the platform to explore the environment with a punishment, consisting of a foot shock through the floor grid. Therefore, on subsequent exposure to the same environment, the mice would avoid or increase their latency before stepping down onto the floor grid. On day 1, for the training test, each mouse was gently placed on the platform and received an electric shock for 3 s when it stepped down and placed all paws on the grid floor. Responsiveness to the punishment in the training test was assessed by animal vocalization or urine production, and only these mice were used for the retention test. Twenty-four hours later, each mouse was placed on the platform again, and the time the mouse remained on the platform was measured, considering 3 min as the upper cut-off time. Following each test, the apparatus was carefully cleaned with Virkon. The tests were carried out between 10:00 a.m. and 1:00 p.m.

### 2.14. Histology and Histochemistry

Full-thickness segments of the distal colon were fixed in 4% paraformaldehyde in 0.1 M phosphate-buffered saline (PBS, pH 7.4) overnight (ON) at 4 °C, dehydrated in graded ethanol series, cleared in xylene, and embedded in paraffin with the cut section transversal to the longitudinal axis. Sections of 5 µm thick segments were cut using a rotary microtome (MR2, Boeckeler Instruments Inc., Tucson, AZ, USA) and collected on slides. The sections were deparaffinized and rehydrated for routine histology and histochemistry. Some sections were stained with hematoxylin–eosin (H&E) to evaluate the tissue integrity, others were submerged for 10 min in 0.1% Toluidine Blue (TB) in 30% ethanol, after filtering, and others were treated for periodic acid–Schiff reagent (PAS) staining. At the end of each staining procedure, the slides were dehydrated, clarified, and mounted in synthetic resin. All of the sections were stained in a single session to minimize artefactual staining differences.

### 2.15. Aβ 1-42 Serum Dosage

Mice were anesthetized through the subcutaneous administration of ketamine–dexmedetomidine mixed solution (80–120 mg/kg + 0.5–1.0 mg/kg, respectively) for euthanasia. During anesthesia, the blood (volume: approximately 0.14 mL) was taken from the mandibular vein and collected in a tube (one sample/mouse). The blood samples were left to coagulate for 2 h at room temperature (RT). After, the tubes were centrifuged at 4000 rpm for 10 min; the obtained sera were transferred into clean tubes and immediately frozen at −80 °C. The dosage of Aβ 1-42 in the serum was determined following the instructions of the specific Elisa kit (Invitrogen, Waltham, MA, USA, Catalog # KMB3441).

### 2.16. Quantitative and Statistical Analysis

All the results are reported as the mean ± S.E.M. Statistical analysis was performed by paired Student’s *t*-test or ANOVA, as appropriate. When ANOVA indicated significant differences, we performed multiple comparisons among groups by the Newman–Keuls post hoc test; *p* < 0.05 was considered significant. Digital images of PAS- and TB-stained structures were acquired with a video-camera-equipped microscope (Eclipse 200; Nikon Instruments, Tokyo, Japan) with a ×10 objective, and the reconstruction of the entire section (2 sections/slide, 2 slides/animal) was performed using appropriate software. Quantitation of PAS and TB staining was performed with the threshold function in ImageJ software (version 1.47, National Institute of Health, Bethesda, ML, USA, http://imagej.net/ij/docs/index.html, accessed on 2 October 2012). In brief, the entire section was commuted to 8-bit grey scale, and the epithelium area of the ascending colon was demarcated. The threshold level for the PAS reaction was selected by dividing the mean grey value of the signal intensity by 1.25, selecting only the stained cells. In TB, to highlight violet staining, the signal range selected was as follows: the minimum value corresponded to the mean grey value of signal intensity, and the maximum value corresponded to the 10% added to the mean value of signal intensity. The data for quantitative analyses of PAS and TB positivity were obtained from the ratio between positive pixels above the threshold and total pixels in each region of interest, expressed as percentage. Statistical analysis on 16S rRNA data was performed by QIIME2 using nonparametric tests. Univariate analyses aimed at evaluating potential differences in the taxonomic composition of the samples were performed by edgeR [37], following centered-log ratio transformation of raw ASV counts. Multi-factor analyses to assess the association of microbial community features with experimental metadata were performed by MaAsLin2 using general linear models [38]. To evaluate global differences in microbiome composition between samples’ groups, the permutational multivariate analysis of variance (PERMANOVA) test was applied to beta-diversity ordination measures to test significance between potential clusters recognized on PCoA plots; significance was determined through 999 permutations.

## 3. Results

### 3.1. Body Weight

The body weight gain showed a regular trend in both female and male mice, regardless of which experimental group they belonged to. At the end of the treatment (age 8 months), their mean weight was 30 to 35% above the baseline weight at 2 months of age, in both genders (Table 1).

### 3.2. Food/Water Intake and Fecal Production

The mean amounts of daily food and water intake during treatment were not significantly different among groups for both male and female mice (Appendix A), confirming that the addition of prebiotics and probiotics to the diet did not influence the food needs of the animals. However, over time, Tg mice showed a modification in intestinal activity, as their stool production, expressed either as weight or as number of fecal pellets, was constantly reduced, and at the end of the treatment (T180) this reduction became significant compared to the other groups of mice, at both 4 and 8 months of age (Figure 1).

### 3.3. Behavioral Tests

#### 3.3.1. Rotarod

The application of the rotarod test at the 4th and 8th months demonstrated that the motor coordination of the mice was unchanged in both sexes and in all groups. All mice improved their performance between the 1st and 3rd trials, with an increase in the latency of the first fall and a reduction in the total number of falls during the trials, at both speeds applied (Appendix A).

#### 3.3.2. Marble Burying

The number of marbles buried by each mouse was low and comparable among all of the groups (Appendix A). This result demonstrated the absence of innate anxious behaviors or the presence of stressors in the environment, which could influence the mice’s performance during other behavioral tests.

#### 3.3.3. Nesting

The evaluation of the mice’s nesting ability highlighted an unequal distribution of the scores among the groups. The highest scores were assigned to the Wt and Wt-T groups (Figure 2A,B), and the lowest to the Tg mice, in whose cages the nesting material (Cocoon) was largely untouched or partially torn up (Figure 2C). The Tg-T mice built recognizable nests and obtained scores comparable to those of the Wt and Wt-T groups (Figure 2D). The statistical analysis revealed a significant difference between the Tg group and the other groups, at both the 4th (T60, Figure 2E) and 8th (T180, Figure 2F) months of age.

#### 3.3.4. Novel Object Recognition (NOR)

The amount of time spent with the novel object and the discrimination index revealed a preference for the novel object by Wt, Wt-T, and Tg-T mice, while Tg mice showed a significant inability to discriminate between the novel object and the previously explored object. These differences in exploratory behavior towards the novel object were recorded at 4 months of life (T60, Figure 3A,B) and confirmed at 8 months of life (Figure 3C,D).

#### 3.3.5. Barnes Maze

This test was applied only at 8 months of age. Over the training days, all of the groups showed a progressive reduction in their latency to find the escape hole, and the first 5 days represented the effective period of learning. However, the Tg mice took significantly longer to reach the escape hole (Figure 4A) and spent less time in the target quadrant (Figure 4B) compared to Wt, Wt-T, and Tg-T mice during the first days. At day 5, all animal groups showed comparable results.

#### 3.3.6. Step-Down Inhibitory Avoidance Test

This test was only applied at 8 months of age. During training, all mice stepped down after a short and comparable latency time. During retention, recall of the punishing experience caused a significant increase in latency before stepping down in Wt, Wt-T, and Tg-T mice. In contrast, Tg mice, during retention, jumped down with a latency that was not significantly different from that shown during training (Figure 5). 

### 3.4. Histology and Histochemistry

#### 3.4.1. H&E Staining

H&E staining of the colonic transversal sections did not show significant differences among the groups in the muscle coat. The submucosa appeared devoid of inflammatory infiltrates in Wt, Wt-T, and Tg-T mice, while grouped lymphocytes were seen in some Tg sections. The mucosa had a normal appearance in all groups; however, in Tg-T mice, the villi were particularly trophic and well developed compared to Tg mice (Appendix A). Quantification of the mucosal area showed no significant differences among groups (personal observation).

#### 3.4.2. PAS and TB Staining

PAS was used to label the mucus present in goblet cells and glandular crypts (Figure 6A,B), and its quantitation showed a significant decrease in Tg mice compared to all of the other groups (Figure 6E). TB, which estimates the acidic component of the mucus, was detected in the majority of secreting cells in all groups of animals (Figure 6C,D), and its quantitation was comparable among the groups (Figure 6F). Briefly, the ratio between the TB- and PAS-stained cells favored increased acidic secretion in Tg mice (Figure 6G).

##### Aβ 1-42 Detection in Serum

The assay of Aβ1-42 in serum blood samples showed the presence of the protein in all groups of mice. However, in Tg mice, the concentration of Aβ1-42 was significantly higher compared to all of the other groups of mice (Figure 7). 

##### Profiling of the Gut Microbiota

The structure (i.e., α- and β-diversity) and composition (i.e., taxonomic profiles) of the gut MB were investigated in fecal samples obtained from Wt (n = 14) and Tg (n = 20) mice, before (Wt-T0, Tg-T0) and after (Wt-T180, Tg-T180) supplementation of the pre/probiotic mixture (Wt-T180-T, Tg-T180-T) over 6 months.

Analysis of α-diversity, assessed through the Chao1 and Shannon diversity indices, revealed no major differences in the Wt-T0 and Tg-T0 groups (Mann–Whitney *p* = 0.22727 [Chao1]; *p* = 0.90387 [Shannon]) (Appendix A). Conversely, an overall lower microbial diversity was observed in both the Wt-T180-T and Tg-T180-T groups compared to their counterparts that were not subjected to the treatment (Appendix A), although such trends were not consistently significant when evaluated with both diversity indices (Mann–Whitney *p* = 0.41359 for Wt, *p* = 0.036542 for Tg [Chao1]; *p* = 0.10789 for Wt, *p* = 0.11194 for Tg [Shannon]).

Potential differences in the gut community profiles of Wt and Tg mice were further explored through β-diversity analysis. The comparison of the overall bacterial community structure through PCoA, using the Bray–Curtis metric, revealed a defined cluster of samples from the Wt-T180-T and Wt-T180 groups compared to those from Wtl-T0; however, no evident clustering patterns were recognized with respect to the treatment (Figure 8). Nevertheless, statistical analysis revealed significant differences in the β-diversity distances of samples from the Wt-180 groups (PERMANOVA *p* = 0.013, F-value: 2.1512, R-squared: 0.16357).

Concerning Tg mice, a defined cluster of samples from the Tg-T180-T and Tg-T180 groups was observed compared to the Tg-T0 baseline samples (Figure 8). Unlike the controls, however, samples from the Tg-T180-T group also exhibited a defined clustering pattern compared to samples from the Tg-T180 group, suggesting a different taxonomic composition of the gut MB, likely associated with the treatment (Figure 8). Statistical analysis revealed significant differences in the β-diversity distances of samples from the Tg-180 groups (PERMANOVA *p* = 0.001, F-value: 3.5338, R-squared: 0.16411).

A comparison of the taxonomic profiles of all experimental groups revealed no major differences at the phylum level, with dominance of *Bacteroidota* and *Firmicutes* (Figure 9). At lower taxonomic levels (i.e., family), univariate analysis revealed that several microbial taxa had a significantly different relative abundance between the treated and untreated groups, mostly among Tg mice (Table 2). Multivariate analysis, using time (T0, T180) and treatment (no treatment, treatment) as covariate adjustments, revealed that *Akkermansiaceae* were significantly reduced in samples from Tg-180 compared to those from Tg-180-T (*p* = 0.00402, FDR *p* = 0.0357); consistently, the genus *Akkermansia* showed a similar trend between the Tg-180 and Tg-180-T groups (*p* = 0.00402, FDR *p* = 0.0342). No significant variations in the relative abundance of *Akkermansiaceae* and *Akkermansia* were observed between samples from Wt-180 and Wt-180-T.

## 4. Discussion

The present study confirms the critical role of the gut MB in the origin of cognitive impairment in our animal model of AD. In fact, the application of behavioral tests that investigate different types of brain performance shows that APPPS1 mice, at the two ages evaluated, suffered from learning and memory deficits but did not present motor disorders or anxiety. Importantly, the deficits were prevented by the treatment with a mixture of prebiotics and probiotics, likely shaping the functions exerted by the gut MB. Furthermore, the treatment was effective in preventing the reduced mucus secretion of the intestinal epithelium and the increase in blood levels of Aβ, which were otherwise observed in untreated Tg mice.

The presence of amyloid deposits in the extracellular space of cerebral tissue is commonly considered to be a hallmark of AD [18,39]. The main brain targets of these deposits are the cortex and the hippocampus, regions that are greatly involved in behavioral skills. Unfortunately, the identification of the clinical signs, which reveal that a neurodegenerative process is underway, is late. As shown by fine instrumental investigations, the time of the diagnosis often coincides with the presence of large amounts of Aβ plaques in the brain, greatly limiting any attempt to prevent or border the damage [39]. Nevertheless, several pharmacological therapies have been attempted [7,24,26], often with scarce and disappointing results. However, all of the researchers agreed that positive effects could be achieved only by starting at the very beginning of the disease, during the so-called MCI (mild cognitive impairment) stage, or even earlier [26,39,40,41]. Moreover, a common belief is emerging according to which acting on peripheral sources of Aβ could be the key to obtaining significant results in efforts to cure the disease [9,39,41].

The present experimental study responds to both requests, as the chosen treatment starts at an early age and targets one of the major producers of Aβ outside the brain, the gut MB [7,9,18,42].

The evaluation of the physiological parameters (appetite, water intake, weight gain) showed no significant differences among the groups, indicating a state of general well-being and rapid adaptation to the gelled water. Nevertheless, the Tg mice showed reduced fecal production compared to all of the other groups, starting from the 4th month of age, which became significant at the 8th month. Intestinal impairment, mostly constipation, has often been described in patients with neurodegenerative diseases (AD, Parkinson’s disease, amyotrophic lateral sclerosis), and this symptomatology far precedes the neurological deficits [43]. Reduced fecal production could depend on several factors, such as neuropathy, intestinal barrier (IB) leakage, changes in the mucus layer’s properties, or dysbiosis, and all of these factors are closely related and influence each other [44,45].

At 8 months of age, the Tg mice showed a significant reduction in total mucin production in the ascending colon, which was not accompanied by a decrease in the acidic component, thus resulting in an imbalance towards a more acidic secretion.

In the mouse colon, a thick and continuous mucus layer covers the epithelium, and its quantity and composition depend on goblet cells, immune cells, and the MB [45,46,47]. Variations in the quantity and quality of the mucus layer affect the MB, as the bacteria adapt to this layer, expressing a range of adhesins, and are equipped with diverse enzymes to break down mucin glycan chains for nutrition [45]. Moreover, a reduced thickness of this layer causes IB dysfunction [24] and enables pathogenic species (such as *E. coli*) to reach the epithelium [45]. The present treatment prevented the decrease in fecal production and mucus secretion. At 8 months, the Tg mice also displayed a significant increase in their plasmatic Aβ1-42 levels, which was prevented by the treatment with pre- and probiotics. The presence of altered mucus secretion with relative loss of IB tightness, along with the changes in the MB found in the Tg mice, may explain the increase in Aβ levels in the bloodstream. Notably, it has been reported that, with age, the blood route plays a major role in determining the Aβ load in the brain [7], compared to the vagal pathway [8].

Profiling of the gut MB revealed that a limited number of taxa had a significant variation in their relative abundance following treatment, mostly among Tg-180 compared to Wt-180 mice.

Interestingly, a significantly decreased abundance of the genus *Akkermansia* was identified in Tg-180 mice compared to the Tg-180-T, Wt-180, and Wt-180-T groups, representing a relevant microbial signature that was likely associated with the treatment. *Akkermansia* is indeed an intestinal symbiont, ranking among mucolytic bacteria (e.g., *Bifidobacterium* spp., *Ruminococcus gnavus*, *Bacteroides thetaiotaomicron*), which are known for their abilities to bind and utilize mucins [48,49]. In recent years, this genus (primarily represented by *A. muciniphila*) has gained increasing attention, since its abundance is closely related to human health (e.g., reinforcement of the mucosal barrier in mice and humans by increasing the mucus layer’s thickness) [50,51]. Likewise, a reduced abundance of *Akkermansia* has been repeatedly associated with a wide range of disorders representing important risk factors for AD, including obesity, type 2 diabetes, IB dysfunction, and other metabolic syndromes [51]. Interestingly, a recent study reported that administration of *A. muciniphila* for six months had a significant effect on the progression of AD in APPPS1 mice, alleviating the reduction in colonic mucus cells and reducing the level of Aβ plaque deposition in the hippocampus and the cerebral cortex. Overall, exposure to *Akkermansia* was associated with significant protective effects against cognitive deficits and amyloid pathology in AD mice [24]. Furthermore, the same study reported that the abundance of *A*. *muciniphila* decreased with age in APPPS1 mice, consistent with another report [15], thus reinforcing the notion that its supplementation could represent a new approach for the prevention and treatment of AD [24]. In this scenario, our results suggest that early supplementation with a symbiotic mixture, composed of a multi-extract of fibers, plant complexes, and Lactobacilli, can indirectly mediate an improvement in the gut barrier’s functions and delay the pathological changes associated with AD, most likely by counteracting the age-related reduction in *Akkermansia* observed in APPPS1 mice. As noted, no significant increase in Lactobacillus spp. was recorded. As such, the symbiotic intake might indirectly shape the balance of the gut microbiota, as well as that of host homeostasis, by influencing various metabolic functions (i.e., metabolome). Additional investigations are needed to provide a mechanistic understanding of the modulation observed in APPPS1 mice following symbiotic administration, and to corroborate this hypothesis.

If peripheral symptomatology can represent a source of discomfort, the signs and symptoms related to brain damage profoundly affect the quality of life of people affected by AD. They consist in important behavioral changes and severe, progressive, and irreversible cognitive deterioration until the loss of all ability to manage daily life. Our animals were thoroughly investigated from this point of view, using tests to monitor their physical and mental performance. When anxiety was measured at 8 months with the marble test, no significant differences emerged among the groups. This result is not consistent with that obtained by Samaey et al. [41]. This incongruence could depend on the use of different Tg strains. However, it should be underlined that these authors evaluated anxiety in young mice (4 months), while our animals were twice as old. Retrospective studies in humans affected by AD have shown that behavioral changes consisting of anxiety, irritability, or depression preceded the cognitive impairment and the diagnosis by many years [52]. Thus, we cannot exclude the possibility that our animals, if tested at 4 months, would have shown anxiety. In agreement with Samaey et al. [41], our animals, both at 4 and 8 months of age, did not show physical weakening when motor coordination was assessed using the rotarod.

At variance, a complex and variegate picture emerged when cognitive tests were applied. All of the tests used explored natural murine behaviors but highlighted different aspects of cerebral function. The NOR test showed that Tg mice were impaired as early as 4 months and worsened by 8 months. This test is considered optimal for exploring the cognitive status of rodents [53]; it is based on the innate exploratory behavior of mice and assesses the recognition memory [35], one of the functions that is constantly and early altered in AD [53]. Again at 4 and 8 months of age, we tested another innate behavior of mice: nesting. The Tg mice already showed great difficulty in building a functional nest at 4 months. Nesting belongs to procedural memory, a form of innate memory that concerns daily life behaviors and survival [41]. The meaning of nesting for mice has been translated to the ADL (activities of daily living) in humans [34], one of the tests that is usually applied when a suspicion of cognitive impairment is raised. Although this assessment is positive in the conclamant stage of AD, subtle differences in instrumental ADL might be present up to 10 years before diagnosis [54]. Nutraceutical treatment was able to significantly improve the mice’s performance, as Tg-T mice achieved values comparable to those of Wt and Wt-T mice in both the NOR and nesting tests.

Another cognitive function that is constantly and early impaired in AD patients is spatial learning and memory, a hippocampus-dependent task [36,55]. Classically, this function is tested using the Morris water maze. However, a well-established alternative is the Barnes maze [36], a land-based rodent test that offers the advantage of being free from the confounding influence of swimming behavior, which is not naturally developed in mice. Initially proposed by Carol Barnes [56] for rats, it was later adapted for mice. Because of its potency to prime memory, it should not be applied twice in the same animal; thus, we tested this maze only once, at 8 months of age. The results obtained were quite intriguing. During the first days of training, Tg mice took significantly longer to learn and orient themselves to reach the escape hole compared to all of the other groups. However, at day 5, all of the animals, regardless of the groups they belonged to, showed comparable performances. Thus, the Barnes test confirmed that Tg mice have a deficit in spatial memory, but that this deficit may recover with training, and that the nutraceuticals are effective in preventing this deficit, as the Tg-T mice did not perform significantly differently from the two Wt groups. In brief, the Barnes maze, while highlighting the presence of an impairment in spatial memory in 8-month-old APPPS1 Tg mice—as expected, since at this age numerous Aβ plaques are present in the hippocampus [27]—also displays that the hippocampus retains the residual capacity to compensate for the damage through behavioral reinforcement.

The last test applied was the step-down inhibitory avoidance test. This measures the ability to memorize a short-term negative experience and recall it when the animal is exposed to the same environment. The Tg mice at 8 months showed a clear deficit in recalling the punishment. The treatment prevented this deficit. Similar results were obtained when probiotics were administered in AD mice [14,26,57,58]. The present study, however, presents some novelties compared to many others: the animals used belonged to both sexes, as differences between females and males have been reported [58,59], and the treatment was extended to a group of Wt mice; the nutraceuticals were administered orally (the common route used in humans) to avoid the trauma of gavage, which could have affected the behavioral evaluations; the pre- and probiotics were administered together, as prebiotics are the metabolic substrates of many members of Firmicutes (Gram+) that have been reported to decrease with age [6,14,16]. Moreover, prebiotics are also the substrate of the Lactobacilli present in our probiotics, and they might sustain their survival. Finally, an extensive and varied battery of behavioral tests was applied, among which the Barnes maze and the nesting test gave very interesting results.

## 5. Conclusions

In conclusion, our data demonstrate that long-term treatment with pre- and probiotics, started early in life, when no signs of the disease were detectable in our mice, was able to prevent or significantly limit relevant biological and behavioral changes and microbial imbalances that were otherwise observed in APPPS1 (untreated) mice; they confirm the existence of an interdependence between the components of the MB–gut–brain axis, strengthening the role of the complex cross-talk occurring along the gut–brain axis [7,18,60,61], and providing further evidence about how modulation of the gut MB might translate into amelioration of AD pathology.

All of this information raises a first question: is it possible to translate this schedule of treatment to humans? We believe that it could be. Eight-month-old mice can be compared to middle-aged humans. At this age, people who will develop AD later in life may present behavioral changes that are difficult to diagnose, such as due to a psychiatric illness or the onset of a neurodegenerative process [18,39,41]. However, it is precisely at this stage, or even earlier, that it would be necessary to make the correct diagnosis and intervene [17,18,39,41]. Interventions, as occur in complex clinical pictures such as AD, must be multidisciplinary; among these, the addition of a constant intake of nutraceuticals to the diet, a treatment without side effects, could be effective and desirable. The next question that needs to be answered is, beyond the very few familiar forms, how to select potential AD patients? It has been reported that people with AD often present dysbiosis even before neurological symptoms become evident [42,62]; therefore, fecal MB screening, a relatively simple and non-invasive procedure, at ages between 40 and 50 years, could be a selection criterion.

## Figures and Tables

**Figure 1 nutrients-16-02381-f001:**
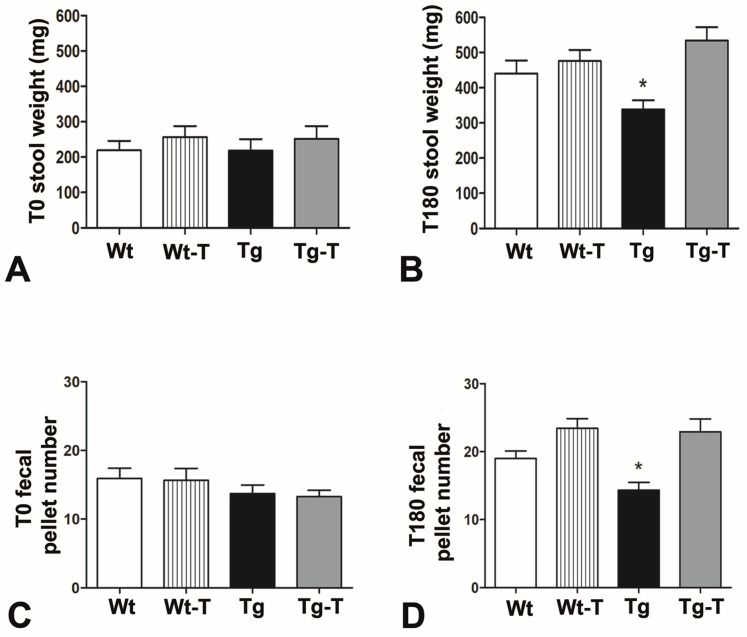
Stool production: (**A**,**C**) At T0 (4 months of age), the stool weight and the fecal pellet numbers were similar among the four groups of mice. (**B**,**D**). At T180 (8 months of age), both the stool weight and the fecal pellet numbers of Tg mice were significantly reduced compared to all of the other groups; * *p* < 0.05, One-way ANOVA, post hoc Newman–Keuls multiple comparison test. Wt n = 13; Wt-T n = 13; Tg n = 15; Tg-T n = 14.

**Figure 2 nutrients-16-02381-f002:**
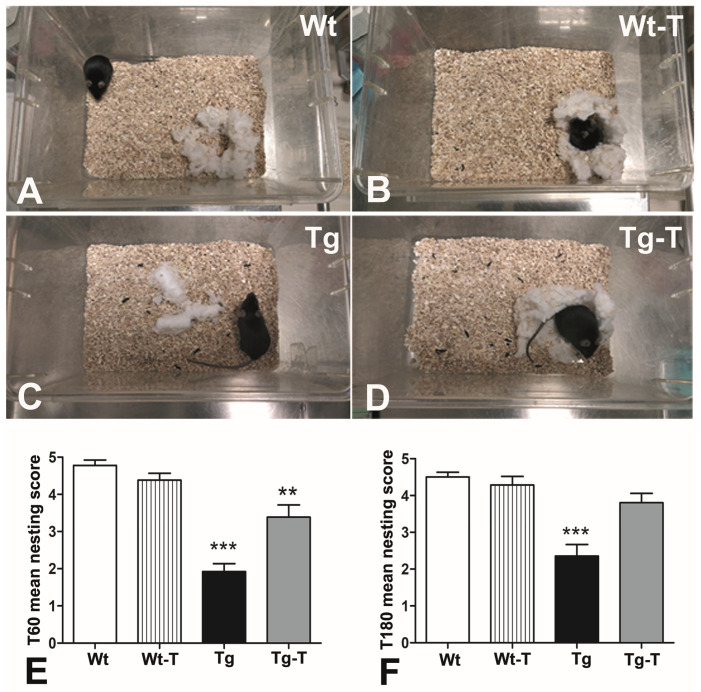
Nesting: Representative images of nests made using a cylinder of cotton (Cocoon). (**A**,**B**) Wt and Wt-T mice; a nest with a central crater and a wall higher than the animal’s body height; score 5. (**C**) Tg mice; no identifiable nest site is seen; the nesting material is threadbare for more than 50%; score 3. (**D**) Tg-T mice; a flat nest is clearly identifiable; score 4. (**E**) The statistical analysis of the scores assigned to the nest quality at T60 showed a significant difference between Tg and all of the other groups, as well as between Tg-T and Wt/Wt-T mice. (**F**) At T180, the performance of Tg mice was significantly worse than all of the other groups. ** *p* < 0.005, *** *p* < 0.0001. One-way ANOVA, post hoc Newman–Keuls multiple comparison test. T60: Wt n = 9; Wt-T n = 13; Tg n = 13; Tg-T n = 13. T180: Wt n = 11; Wt-T n = 13; Tg n = 12; Tg-T n = 13.

**Figure 3 nutrients-16-02381-f003:**
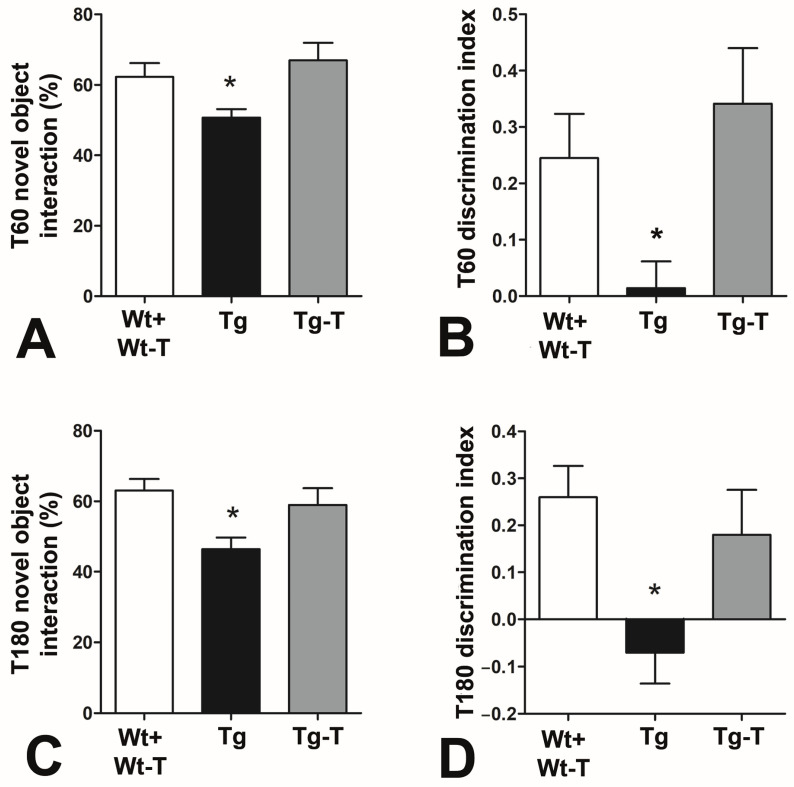
Novel object recognition: (**A**,**C**) Percentage of time spent exploring the novel object with respect to the total time. (**B**,**D**) Discrimination index (DI) at T60 and T180. Both parameters were significantly reduced in Tg mice at both times of treatment. * *p* < 0.05 vs. the other groups. One-way ANOVA, post hoc Newman–Keuls multiple comparison test. T60: Wt +Wt-T n = 9; Tg n = 14; Tg-T n = 8. T180: Wt +Wt-T n = 14; Tg n = 14; Tg-T n = 11.

**Figure 4 nutrients-16-02381-f004:**
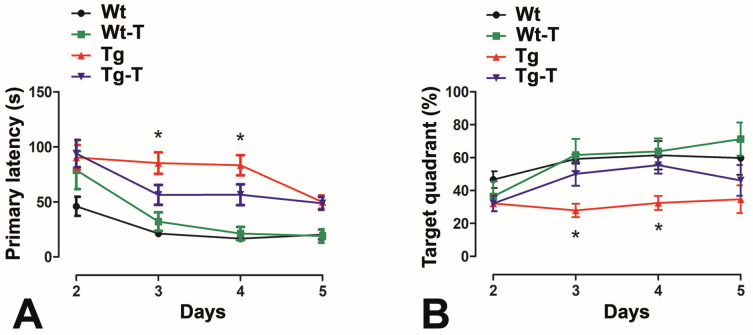
Barnes maze test: The graphs show the analysis of the mice’s performances during the first 5 days of the test, which represent the effective period of learning. (**A**) The primary latency, i.e., the time (s) to reach the escape hole, was significantly higher in Tg mice at the 3rd and 4th days. (**B**) The time spent by the mice in the target quadrant, expressed as % of the total time, was significantly reduced for Tg mice at the 3rd and 4th days. * *p* < 0.05 vs. the other groups. One-way ANOVA, post hoc Newman–Keuls multiple comparison test. Wt n = 12; Wt-T n = 16; Tg n = 20; Tg-T n = 14.

**Figure 5 nutrients-16-02381-f005:**
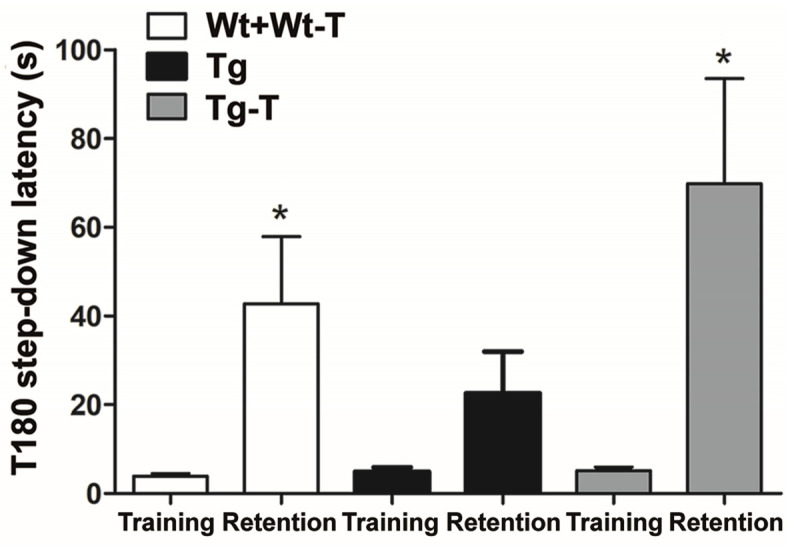
Step-down inhibitory avoidance test: The graph shows the latency (s) before stepping down during the training phase (with punishment) and during the retention phase (without punishment). Tg mice showed a significant reduction in latency during the retention phase; * *p* < 0.05 vs. the training phase, paired Student’s *t*-test. Wt + Wt-T n = 16; Tg n = 12; Tg-T n = 12.

**Figure 6 nutrients-16-02381-f006:**
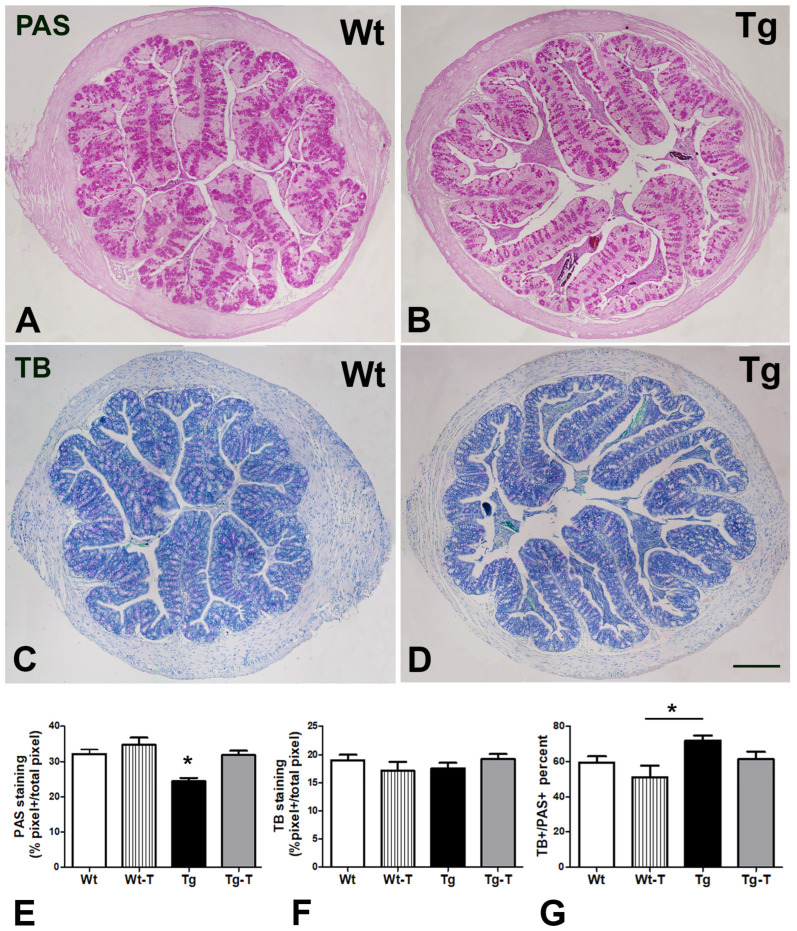
Periodic acid–Schiff (PAS) and Toluidine Blue (TB) staining: (**A**,**B**) The PAS staining, in pink, is distributed in the goblet cells along the villi and in the cells of the crypts. (**C**,**D**) TB staining highlights the acidic component of the mucous in violet. Bar = 200 μm. Quantitation of both dyes showed a significant decrease in the PAS/pink component (**E**) in Tg mice, with no significant change for the BT/violet component (**F**). The ratio between violet TB staining and total pink PAS staining was significantly higher in the Tg mice (**G**). * *p* < 0.05 vs. the other groups. One-way ANOVA, post hoc Newman–Keuls multiple comparison test. Wt n = 9; Wt-T n = 8; Tg n = 9; Tg-T n = 9.

**Figure 7 nutrients-16-02381-f007:**
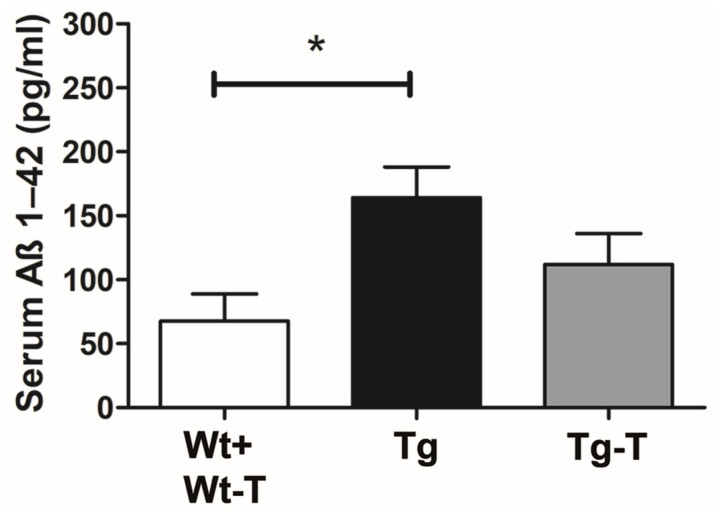
Aβ 1-42 serum dosage: The dosage of Aβ 1-42 protein in the serum, expressed in pg/mL, showed a significant increase in Tg mice compared to Wt and Wt-T mice. The results obtained in the Tg-T mice did not differ from those of the other groups. * *p* < 0.05. One-way ANOVA, post hoc Newman–Keuls multiple comparison test. Wt + Wt-T n = 14; Tg n = 16; Tg-T n = 15.

**Figure 8 nutrients-16-02381-f008:**
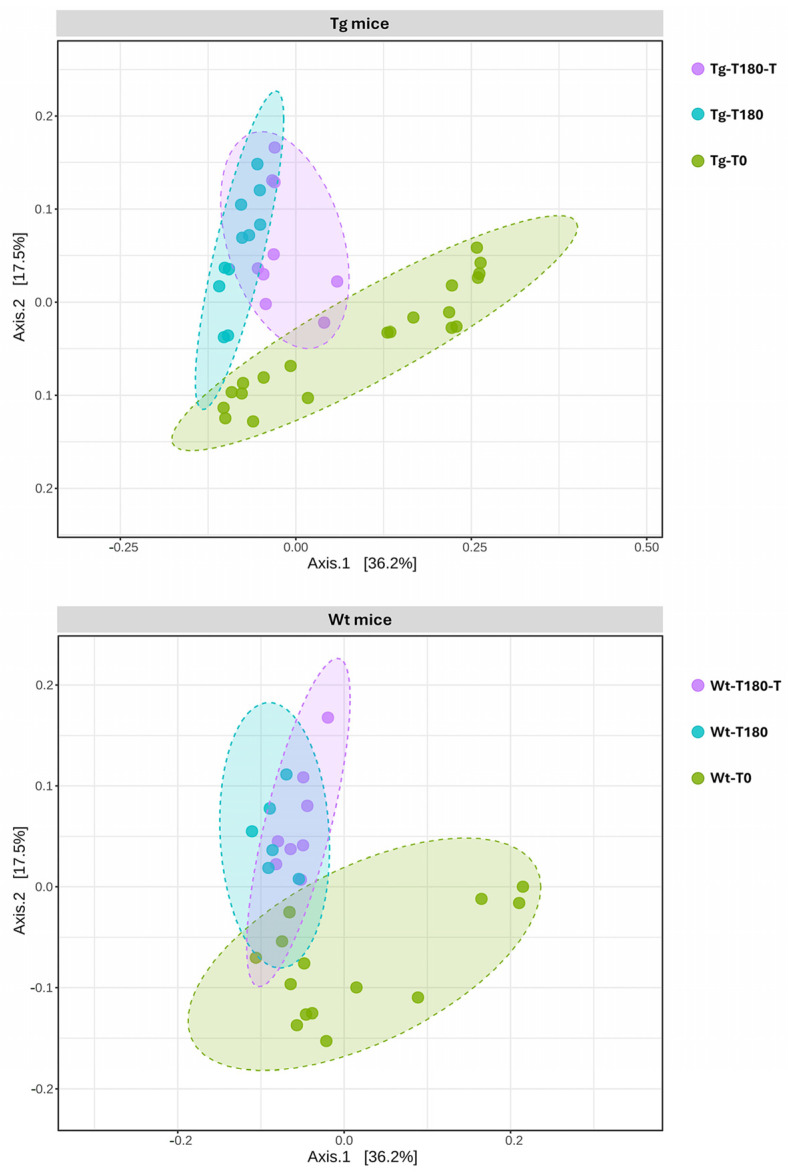
Analysis of β-diversity: Principal coordinates analysis (Bray–Curtis β-diversity metric) of samples obtained from wild-type (Wt) and transgenic APPPS1 (Tg) mice, before (Wt-T0, Tg-T0) and after (Wt-T180, Tg-T180) supplementation of the pre/probiotic (Wt-T180-T, Tg-T180-T) mixture.

**Figure 9 nutrients-16-02381-f009:**
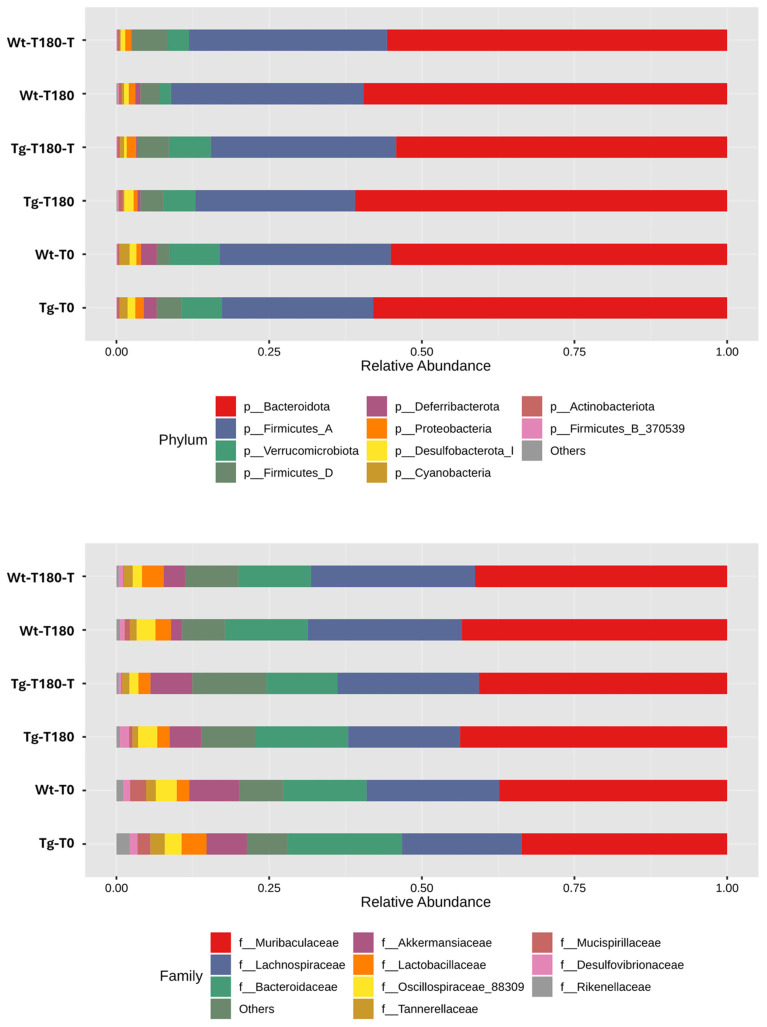
Taxonomic profiling: Breakdown of taxonomic profiles (top 10 taxa) at phylum-level (**upper** panel) and family-level (**lower** panel) indices in samples obtained from wild-type (Wt) and transgenic APPPS1 (Tg) mice, before (Wt-T0, Tg-T0) and after (Wt-T180, Tg-T180) supplementation of the pre/probiotic (Wt-T180-T, Tg-T180-T); relative abundance values are also shown (*y*-axis).

**Table 1 nutrients-16-02381-t001:** Body weight gain.

Body Weight Gain				
**2** **nd month of Life**	**Wt**	**Wt-T**	**Tg**	**Tg-T**
Male	23.91 ± 0.32 g	22.54 ± 0.48 g	21.72 ± 0.10 g	22.02 ± 0.6 g
Female	18 ± 0.36 g	19.35 ± 0.31 g	17.68 ± 0.36 g	17.69 ± 0.59 g
**8** **th month of Life**	**Wt**	**Wt-T**	**Tg**	**Tg-T**
Male	32.11 ± 1.17 g	30.52 ± 0.35 g	29.9 ± 0.92 g	30.59 ± 0.44 g
Female	25.47 ± 1.5 g	25.65 ± 0.89 g	24.65 ± 0.4 g	23.92 ± 0.24 g

**Table 2 nutrients-16-02381-t002:** Overview of microbial taxa showing statistically significant variations in their relative abundance between treated and untreated groups of wild-type (Wt-180) and Tg (Tg-180) mice. The arrows indicated an abundance reduction in treated vs. untreated groups.

**Wt-180 Groups**						
**Phylum**	**Class**	**Order**	**Family**	**P**	**FDR**	**Trend Observed in Treated vs. Untreated**
*Micutes*	*Bacilli*	*Erysipelotrichales*	*Coprobacillaceae*	1.056 × 10^−4^	0.0044351	
*Bacteroidota*	*Bacteroidia*	*Bacteroidales*	*Marinifilaceae*	0.0021527	0.045206	↓
**Tg-180 Groups**						
**Phylum**	**Class**	**Order**	**Family**	**P**	**FDR**	**Trend Observed in Treated vs. Untreated**
*Patescibacteria*	*Saccharimonadia*	*Saccharimonadales*	*Nanosyncoccaceae*	3.9799 × 10^−7^	1.6715 × 10^−5^	↓
*Firmicutes*	*Bacilli*	*RF39*	*UBA660*	7.783 × 10^−5^	8.1722 × 10^−4^	↓
*Firmicutes*	*Clostridia*	*Oscillospirales*	*Acutalibacteraceae*	2.7145 × 10^−4^	0.0022802	↓
*Firmicutes*	*Clostridia*	*Christensenellales*	*Borkfalkiaceae*	0.0017769	0.010662	
*Firmicutes*	*Clostridia*	*Christensenellales*	*CAG−74*	1.9848 × 10^−5^	4.1681 × 10^−4^	↓
*Desulfobacterota*	*Desulfovibrionia*	*Desulfovibrionales*	*Desulfovibrionace*	0.0041844	0.021968	↓
*Proteobacteria*	*Alphaproteobacteria*	*D84*	*Rs_D84*	0.0047901	0.022354	

## Data Availability

Data are available upon request from the authors.

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
