# Peer review of "Amelioration of Serum Aβ Levels and Cognitive Impairment in APPPS1 Transgenic Mice Following Symbiotic Administration"

_nutrients, 2024, doi:10.3390/nu16152381_

Round 1
Reviewer 1 Report
Comments and Suggestions for Authors
The manuscript by Chiara Traini et al. is very interesting and informative. The results reinforced the notion that gut-brain axis has a critical role of in the genesis of cognitive impairment and that modulation of the gut microbiota can ameliorate AD symptomatology. I only have two suggestions:
1, for all the tables, please revise and put them in three-line format.
2, limitations of the present study and future research directions should be discussed. For example, how to understand the changes of the gut microbiota after symbiotic intake. I noticed that there was no increase of Lactobacillus spp. Why? Besides, have the authors checked the changes of the metabolic activity of the gut microbiota?
Author Response
Reviewer 1
The manuscript by Chiara Traini et al. is very interesting and informative. The results reinforced the notion that gut-brain axis has a critical role of in the genesis of cognitive impairment and that modulation of the gut microbiota can ameliorate AD symptomatology. I only have two suggestions:
We thank the referee for the nice comments she(he) made on our study
1, for all the tables, please revise and put them in three-line format.
We changed the tables in the format requested
2, limitations of the present study and future research directions should be discussed. For example, how to understand the changes of the gut microbiota after symbiotic intake. I noticed that there was no increase of Lactobacillus spp. Why? Besides, have the authors checked the changes of the metabolic activity of the gut microbiota?
We thank the Reviewer for the comments. As noticed, no significant increase in Lactobacillus spp. was recorded. As such, the symbiotic intake might indirectly shape the balance of the gut microbiota, as well as the of host homeostasis, by influencing various metabolic functions (i.e. metabolome). Additional investigations are needed to provide a mechanistic understanding of the modulation observed in APPPS1 mice following symbiotic administration and to corroborate this hypothesis. A comment has been added to the text to clarify this point.
Reviewer 2 Report
Comments and Suggestions for Authors
Transgenic Mice following Symbiotic Administration", by Chiara Traini and co-workers, is an interesting study in which the authors analyze the relationships between the accumulation of protein amyloid-14 ß (Aβ), a histopathological hallmark of Alzheimer's disease, and the microbiota and dysbiostic states. Es decir, el trabajo se centra en el estudio del gut-brain axis.
The authors start from the hypothesis that maintaining or restoring the microbiota eubiosis is essential to control Aß production and deposition in the brain, since recent studies that have shown que gut microbiota produces Aß and dysbiostic states have been identified in Alzheimer's disease patients and animal models of the disease. The authors use as a model APPPS1 males and female mice and evaluate the cognitive performances, mucus secretion, Aβ serum concentration, and microbiota composition after receiving treatment with a mixture of probiotics and prebiotics (symbiotic). Study findings show that treatment was able to prevent the memory deficits, the reduced mucus secretion, the increased Aβ blood levels and the imbalance in the gut microbiota of the APPPS1 mice. Therefore, they show that the gut-brain axis has a critical role of in the genesis of cognitive impairment and that modulation of the gut microbiota can ameliorate Alzheimer's disease symptomatology.
The manuscript is easy to read, it is well planned and provides new data that is worth taking into consideration, the materials and techniques of the study are adequate and are described in detail and in a way that allows its reproduction. The results are clearly written and are no speculative.
However, it would be interesting for authors to use the conditional more in the discussion because, although the results are very interesting, great caution must be exercised when trying to transfer the results to humans.
I thank the authors and the editor for giving me the opportunity to read this interesting article.
Author Response
Reviewer 2
Transgenic Mice following Symbiotic Administration", by Chiara Traini and co-workers, is an interesting study in which the authors analyze the relationships between the accumulation of protein amyloid-14 ß (Aβ), a histopathological hallmark of Alzheimer's disease, and the microbiota and dysbiostic states. Es decir, el trabajo se centra en el estudio del gut-brain axis.
The authors start from the hypothesis that maintaining or restoring the microbiota eubiosis is essential to control Aß production and deposition in the brain, since recent studies that have shown que gut microbiota produces Aß and dysbiostic states have been identified in Alzheimer's disease patients and animal models of the disease. The authors use as a model APPPS1 males and female mice and evaluate the cognitive performances, mucus secretion, Aβ serum concentration, and microbiota composition after receiving treatment with a mixture of probiotics and prebiotics (symbiotic). Study findings show that treatment was able to prevent the memory deficits, the reduced mucus secretion, the increased Aβ blood levels and the imbalance in the gut microbiota of the APPPS1 mice. Therefore, they show that the gut-brain axis has a critical role of in the genesis of cognitive impairment and that modulation of the gut microbiota can ameliorate Alzheimer's disease symptomatology.
The manuscript is easy to read, it is well planned and provides new data that is worth taking into consideration, the materials and techniques of the study are adequate and are described in detail and in a way that allows its reproduction. The results are clearly written and are no speculative.
However, it would be interesting for authors to use the conditional more in the discussion because, although the results are very interesting, great caution must be exercised when trying to transfer the results to humans.
I thank the authors and the editor for giving me the opportunity to read this interesting article.
First of all, we wish to thank the referee for her (his) words of appreciation of our work. It was a very demanding study; thus, these words are particularly appreciated.
As the reviewer rightly suggested, since we studied animals, precautions need to be taken when transferring these findings to humans. In this regard, some changes have been made to the discussion.